# A Novel Family with Demyelinating Charcot–Marie–Tooth Disease Caused by a Mutation in the PMP2 Gene: A Case Series of Nine Patients and a Brief Review of the Literature

**DOI:** 10.3390/children10050901

**Published:** 2023-05-19

**Authors:** Margherita Baga, Susanna Rizzi, Carlotta Spagnoli, Daniele Frattini, Francesco Pisani, Carlo Fusco

**Affiliations:** 1Child Neurology and Psychiatry Unit, Department of Pediatrics, AUSL-IRCCS di Reggio Emilia, 42100 Reggio Emilia, Italy; 2Child Neuropsychiatric Unit, Human Neuroscience Department, Sapienza University of Rome, 00100 Rome, Italy

**Keywords:** Charcot–Marie–Tooth, demyelinating, PMP2 gene, anticipation

## Abstract

Introduction: Charcot–Marie–Tooth (CMT) is a group of inherited peripheral neuropathies characterized by wide genotypic and phenotypic variability. The onset is typically in childhood, and the most frequent clinical manifestations are predominantly distal muscle weakness, hypoesthesia, foot deformity (pes cavus) and areflexia. In the long term, complications such as muscle-tendon retractions, extremity deformities, muscle atrophy and pain may occur. Among CMT1, demyelinating and autosomal dominant forms, CMT1G is determined by mutations in the PMP2 myelin protein. Results: Starting from the index case, we performed a clinical, electrophysiological, neuroradiological and genetic evaluation of all family members for three generations; we identified p.Ile50del in PMP2 in all the nine affected members. They presented a typical clinical phenotype, with childhood-onset variable severity between generations and a chronic demyelinating sensory-motor polyneuropathy on the electrophysiologic examination; the progression was slow to very slow and predominant in the lower limbs. Our study reports a relatively large sample of patients, members of the same family, with CMT1G by PMP2, which is a rare form of demyelinating CMT, highlighting the genetic variability of the CMT family instead of the overlapping clinical phenotypes within demyelinating forms. To date, only supportive and preventive measures for the most severe complications are available; therefore, we believe that early diagnosis (clinical, electrophysiological and genetic) allows access to specialist follow-up and therapies, thereby improving the quality of life of patients.

## 1. Introduction

Charcot–Marie–Tooth (CMT) disease represents a genetically and phenotypically highly heterogeneous group of inherited peripheral neuropathies. CMT is classified into the demyelinating type (CMT1) with slowed nerve conduction velocity (NCV) below 35 m/s, axonal type (CMT2) with normal or slightly reduced NCV (>45 m/s), and intermediate type (I-CMT) with signs of both demyelination and axonal degeneration and NCVs between 35 and 45 m/s.

The exponential increase in the number of known genes associated with CMT images, possibly mainly due to exome sequencing (WES) techniques, has made traditional classification systems increasingly complex, leading since 2018 to the revision of these systems and classification based on causative gene alterations [1]) Within each category (CMT1, CMT2, etc.), the association with a specific gene is indicated by a letter of the alphabet (e.g., CMT1A, CMT1B, etc.).

CMTs today represent a broad genetic and clinical spectrum; it is not uncommon to find several mutations in the same gene associated with different phenotypes; likewise, it is possible for the involvement of different genes to be associated with the same clinical phenotype.

Genetic variability manifests itself both in different inheritance patterns (autosomal dominant, autosomal recessive and X-linked) and in distinct electroneurophysiological classes (demyelinating, axonal and dominant intermediate). Autosomal dominant forms are the most frequent; a few sporadic cases related to de novo mutations have been described (less than 10% of CMT1A) [2]. Currently, more than 90 different pathogenic mutations are reported in the literature [3] involving protein synthesis and post-transcriptional regulation processes, intracellular transport, or mitochondrial function. [2] 90% of cases are associated with variations in four genes: PMP22, MPZ, GJB1 and MFN2, and about 50% of CMT cases are associated with duplication/deletion of the PMP22 gene [4]. To date, the genetic cause is still unknown in 20–30% of hereditary neuropathies, but this percentage is gradually decreasing due to new molecular techniques [5]. CMT1, a demyelinating type, accounts for 40–50% of all CMT patients [6]. Myelin sheath is composed of a high fraction of lipids and myelin-specific proteins including myelin basic protein (MBP), myelin protein zero (MPZ), peripheral myelin protein 22 (PMP22) and peripheral myelin protein 2 (PMP2). Any slight alteration in its protein and lipid components alters the integrity of its structure and, thus, the axonal lining, resulting in reduced nerve conduction velocities [6]. The most frequent cause of CMT1 is alterations in the PMP22 gene, resulting more frequently in CMT1A and less often in HNPP and CMT1E, followed by MPZ mutations that lead to CMT1B. MBP has not yet been implicated in CMT. PMP2 has recently emerged as a novel, rare cause of the dominant CMT1 [6,7,8,9,10]. PMP2 is a small 14 kDa protein, located on the cytoplasmic side of compact myelin, belonging to the fatty acid binding protein family (FABP) and is characterized by an antiparallel β-barrel structure with an α-helical cap [6,10]. Its function in the peripheral nervous system remains partially unclear, but its role in lipidic homeostasis is suggested in the literature. Stettner et al. also suggested that PMP2 plays a role in the remyelination process of the injured peripheral nervous system [11].

The typical clinical spectrum is characterized by distal weakness, sensory loss, foot deformities and absence of reflexes. In the CMT1 forms reported in the literature, distal hyposthenia of the feet and hands progressively appears along with muscle atrophy. Slowly progressive sensory involvement, with loss of proprioception and vibratory sensitivity, is possible. In young adulthood, there is atrophy of the hand and foot muscles to a greater degree, possible hypertrophy of the nerves, and palpable and kyphosis-scoliosis-like deformities of the spine. The evolution is benign in most cases, and cases of loss of ambulation have been described. Worsening during pregnancy has also been reported.

Here, we report clinical and electrophysiological data from all the members of three generations of a CMT family harboring a heterozygous deletion c.147_149delTAT in the PMP2 gene, which leads to an isoleucine loss at codon 50 (p.Ile50del) that segregates with the disease, as previously described in the literature by Gerlodi et al. [10].

## 2. Results

The family in this case was referred for demyelinating motor sensory neuropathy with autosomal dominant transmission. The index case (IV-1) was a 15-year-old boy born at term by cesarean delivery for placenta previa, following a normal pregnancy (Figure 1). Motor and mental development were normal, as were neurological examinations up to the age of 3 years when high-arched feet and toe walking followed by mild distal tremor were noted. Neurological evaluation at the age of 15 years showed bilateral pes cavus, abolished deep tendon reflexes and mild lower limb distal weakness; postural tremor persisted, which was essentially unchanged over the years, worsening with movement. No pyramidal signs or cranial nerve impairments were observed. Neurophysiological data revealed diffuse and symmetric slowing of both motor and sensory nerve conduction velocity (NCV values between 25 and 30 m/s in nerves analyzed), compatible with sensory-motor demyelinating polyneuropathy. The patient had two sisters: one of them, 23 years old, showed no signs or symptoms of peripheral neuropathy, while the other (IV-2) was a girl of 13 years of age and she presented with regular psychomotor development, early onset of pes cavus, areflexia, mild distal postural tremor and sensory impairment with lower limb dysesthesia that appeared during adolescence. The electrophysiological study was consistent with sensory-motor demyelinating polyneuropathy of the four limbs, with motor velocities values between 25 and 30 m/s and undetectable sensory potentials. Their father (III-1) was a man of 55 years, who presented with foot deformities (pes cavus and hammer toes), areflexia and lower-extremity weakness, particularly ankle dorsiflexion. Similarly, the proband’s grandmother (II-1), 95 years old, showed pes cavus, areflexia and distal lower limb atrophy; she was still able to walk independently, and no severe neuromuscular or orthopedic issues were observed. A neurophysiological study showed NCV of motor nerves with values less than 20 m/s and undetectable sensitive potentials, confirming sensory-motor demyelinating chronic polyneuropathy with predominant sensory involvement. No previous neurological evaluation was available. Medical history revealed that the grandmother’s mother (I-1) and brother (II-2) also showed clinical features of neuropathy with pes cavus and gait disturbance, but no extensive clinical, neurophysiological and genetic data were available. The grandmother’s brother had three daughters: one of them (III-2), 54 years old, presented early onset pes cavus and areflexia and no muscular wasting despite severe distal weakness. The electrophysiological data revealed reduced motor velocities (20 m/s) and absent sensitive potentials, consistent with sensory-motor demyelinating chronic polyneuropathy with predominant sensory involvement and greater involvement of the lower limbs. The second sister (III-3), 57 years old, reported early onset foot deformities, areflexia and motor impairment during childhood, with frequent falls. She showed severe distal weakness and thenar eminence hypotrophy associated with paresthesia. The younger sister (III-4), who was 49 years old, showed only mild foot deformities with pes cavus and areflexia; in adulthood, she began to suffer from mild sensory impairment characterized by hypoesthesia and scoliosis. Unfortunately, their electrophysiological data were not available.

Patient III-4 had two sons with similar clinical phenotypes: the senior one (IV-3) presented with delayed walking at the age of 20 months with frequent falls, mild high arched feet, areflexia, distal tremor affecting the arms and legs and hand, ankle and toe dorsiflexion weakness, and mild atrophy of the anterior compartments of the lower leg and feet; the second one (IV-4) showed independent toe walking at 18 months with rapid progression and appearance of bilateral pes cavus, absence of deep tendon reflexes, gait impairment since the age of 2–3 years, and distal weakness with walking difficulties. Both of them showed a distal postural tremor, worsening with movements, and unchanged over time. For both of them, the NCV studies showed the presence of predominantly demyelinating sensory-motor polyneuropathy with diffuse and symmetric slowing motor nerve conduction velocities (NCV between 25 m/s and 30 m/s) and undetectable sensory potential of the sural and median nerves. With the exception of one patient (II-1), all performed auditory evoked potentials, which were negative, and a neuroradiological brain and lumbosacral study with magnetic resonance imaging that showed no significant abnormalities; only in one case (III-4) was a picture of scoliosis described. The probands were found to be negative for duplication or deletion in the PMP22 gene. Subsequently, a targeted next-generation sequencing (NGS) panel evaluating 157 genes associated with hereditary neuropathy was performed, revealing a deletion c.147_149delTAT in the PMP2 gene, which led to an isoleucine loss at codon 50 (p.Ile50del). Segregation analysis confirmed the presence of this deletion in all the affected family members. The main clinical and instrumental data are summarized in Table 1.

## 3. Discussion

Peripheral myelin protein 2 (PMP2) is a small protein of 132 amino acids located on the cytoplasmatic side of the peripheral nervous system (PNS) compact myelin. Currently, six PMP2 mutations are reported in the literature that are clinically correlated with demyelinating neuropathy (CMT1G); the common molecular mechanism seems to alter the function of the protein PMP2, keeping its structure preserved [10,12].

Although CMT is described as a disease of the peripheral nervous system, CNS involvement is reported in the literature in several CMT patients by variations in the MFN2, PMP22, GJB1, GDAP, MORC-2, NDRG1 and NEFL genes. CNS abnormalities can involve both supratentorial and infratentorial areas; both white matter and grey matter abnormalities, and cerebellar abnormalities are also possible. In particular, in the context of demyelinating CMT1, a reduction in bank substances with signs of diffuse hypomyelination with U-fiber sparing has previously been described in several adult patients with CMT1A and HNPP [13]. In all the cases described here, a possible CNS involvement was ruled out not only by clinical evaluation but also by evoked potentials and neuroradiological study.

Recently, our variant has been described in four members of a family, associated with clinical and electrophysiological features compatible with a classical CMT1 phenotype with delayed acquisition of motor milestones, areflexia, pes equino, toe walking, frequent falls, distal weakness, muscle hypotrophy and reduced nerve conduction velocities, overlapping with the clinical features described in other cases of CMT1G reported in the literature [6,7,8,9,10,14]. Here, we report the same variant described by Geroldi et al. in our index case; this deletion segregates in eight affected members of a three-generation CMT1 family. Our patients also reported two other family members with gait disturbance and likely distal leg deformity, but it was not possible to extend their clinical and genetic analysis. The clinical features of our family are similar to those previously described for the variant (p.ile50del) by Gerlodi et al. However, the electroneurophysiological data in our sample revealed different pictures of sensory-motor neuropathies, with prevalent and early sensory involvement (in most cases, in fact, we found reduced motor conduction velocities of less than 35 m/s and absence of potentials on the sensory nerves investigated), as well as in younger patients. As in the majority of cases, our case also revealed that the lower limbs are more involved than the upper limbs [15]. Sensory nerve involvement is often less severe than motor nerve involvement; it manifests itself with reduced tactile and vibratory sensitivity or reduced proprioception, which contributes to the onset of walking disorders. The involvement of the upper extremities leads to a progressive hypotrophy of the intrinsic musculature with possible contractures in flexion of the fingers. Scoliosis, described only in one patient of our family, is reported in the literature in 30–50% of cases, usually with kyphoscoliosis [16].

An interesting clinical feature in our patients is the presence of early distal tremor, affecting the both lower and upper limb extremities, in younger patients, which is absent in parents. Previously, Hong described adult-onset distal tremor in patients with CMT1G [8]. This intrafamilial phenotypic variability could suggest the phenomenon of disease anticipation in our sample. The same genetic mutation led to variable expression within the phenotype, with variation in clinical features and progressively earlier ages of onset in successive generations.

Although the pathogenetic mechanisms of the phenomenon of disease anticipation are still unknown, the hypothesis is that anticipation and intra-family phenotypic variability could be due to the expression of unknown modulator genes or environmental factors, or both [15,17]. In the context of CMT, it is known that the reduction in the expression of connectin 32, which underlies x-linked CMT, may be the result of either an alteration of transcription factors or a mutation of noncoding (P2-promotor) genes [18]. The same phenomenon of anticipation associated with a progressive increase in clinical severity has also been described in a subgroup of CMT1A patients [19].

## 4. Conclusions

Our study describes the clinical features of a family affected by CMT1G from PMP2. The reported cases belong to three different generations, giving us the possibility to indirectly observe the evolution of the disease over time and to hypothesize, despite the described intra-family phenotypic variability and the phenomenon of disease anticipation in the younger generation, which is a benign course of the disease. Interestingly, in the adult affected members of the family, a slowly progressive clinical course is observed, with a predominant sensory involvement associated with muscle atrophy almost exclusively of the intrinsic muscles of the hands and feet, which to date has not led to severe disabilities and without severe disruption of their quality of life. In the proband, sensory nerves are involved early but less compromised; this could be correlated with an early stage of the disease, as described in other studies [6,10]. Continual follow-ups of new generations appear necessary to monitor the disease and to confirm the overall benign trend. To date, only supportive and preventive measures for the most severe complications are available; therefore, the possibility of an early diagnosis (clinical, electrophysiological and genetic) allows access to specialist follow-up and therapies, thereby improving the quality of life of patients.

## Figures and Tables

**Figure 1 children-10-00901-f001:**
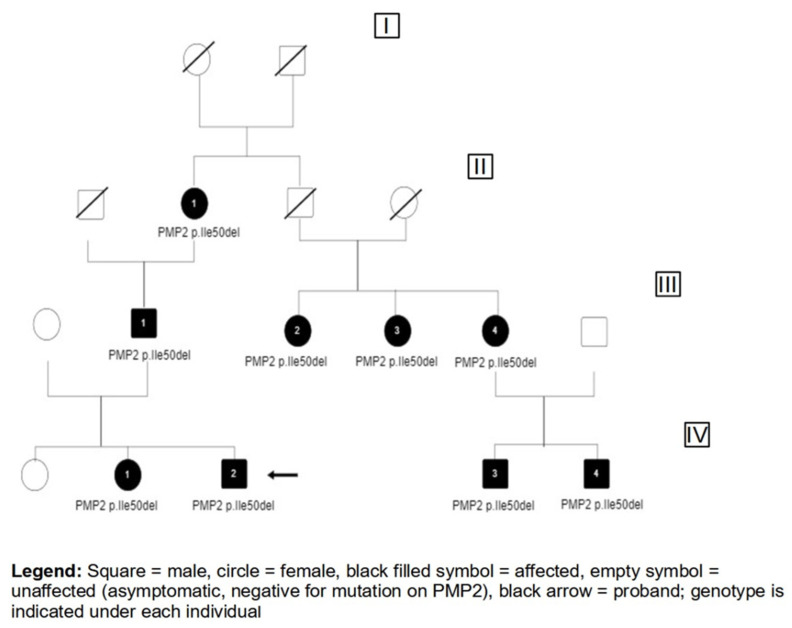
The figure represents the genealogical tree of the family described. For each generation, the subjects are numbered in sequential order; their clinical situation is described in the text.

**Table 1 children-10-00901-t001:** Main clinical and instrumental data; legend: MRI = magnetic resonance imaging, AEP = auditory evoked potential, AD = autosomal dominant, AR = autosomal recessive.

ID Patient	M/F	Age at Onset (Years)	Age at Genetic Diagnosis (Years)	Signs and Symptoms	ENG	MRI	AEP	Genotipic	AD/AR/X-LINKED	Method	Reported in Literature
IV-2	M	2	16	bilateral pes cavus, abolished deep tendon reflex, mild lower limbs distal weakness, postural tremor	diffuse and symmetric slowing motor and sensory nerve conduction velocities (NCV: 25–30 m/s)	normal	normal	c.147_149delTAT of gene PMP2	AD	targeted NGS	YES
IV-1	F	5	13	bilateral pes cavus, abolished deep tendon reflex, mild lower limbs distal weakness, postural tremor, sensitive impairment with lower limbs dysesthesia	symmetric slowing motor nerve conduction velocities 25–30 m/s, undetectable sensory potentials	normal	normal	c.147_149delTAT of gene PMP2	AD	direct Sanger sequencing	YES
III-1	M	unknown	55	bilateral pes cavus and hammer toes, areflexia and lower extremity weakness	motor nerve conduction velocities < 20 m/s, undetectable sensitive potentials	normal	normal	c.147_149delTAT of gene PMP2	AD	direct Sanger sequencing	YES
IV-4	M	7	8	bilateral pes cavus, absence of deep tendon reflexes, gait impairment and distal weakness with walking difficulties. Distal postural tremor, worsening with movements. Back pain	symmetric slowing motor nerve conduction velocities (NCV: 25–30 m/s), undetectable sensory potentials	normal	normal	c.147_149delTAT of gene PMP2	AD	direct Sanger sequencing	YES
IV-3	M	2	13	mild high arched feet, areflexia, distal postural tremor worsening with movements, mild muscular atrophy of the anterior compartments of the lower legs and feet	symmetric slowing motor nerve conduction velocities (NCV: 25–30 m/s), undetectable sensory potentials	normal	normal	c.147_149delTAT of gene PMP2	AD	direct Sanger sequencing	YES
III-3	F	unknown	54	bilateral pes cavus with hammer toes, areflexia, severe distal weakness and thenar eminence hypotrophy associated with paresthesia	not available	normal	normal	c.147_149delTAT of gene PMP2	AD	direct Sanger sequencing	YES
III-2	F	unknown	57	bilateral pes cavus, areflexia, severe distal weakness	not available	normal	normal	c.147_149delTAT of gene PMP2	AD	direct Sanger sequencing	YES
III-4	F	unknown	49	pes cavus, areflexia, distal hypoesthesia and scoliosis	not available	normal	normal, scoliosis	c.147_149delTAT of gene PMP2	AD	direct Sanger sequencing	YES
II-1	F	unknown	96	bilateral pes cavus, areflexia, distal lower limb atrophy	motor nerve conduction velocities < 20 m/s, undetectable sensitive potentials	not performed	not performed	c.147_149delTAT of gene PMP2	AD	direct Sanger sequencing	YES

## Data Availability

No new data were created or analyzed in this study. Data sharing is not applicable to this article.

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
