# Peer review of "A Novel Family with Demyelinating Charcot–Marie–Tooth Disease Caused by a Mutation in the PMP2 Gene: A Case Series of Nine Patients and a Brief Review of the Literature"

_children, 2023, doi:10.3390/children10050901_

Round 1
Reviewer 1 Report (Previous Reviewer 4)
Although the authors have taken on board the suggestions from previous reviewers there are still several points that need to be improved.
There are several points in the case reports that need to be addressed:
e.g NCV in the results was not explained, sensitive demyelinating neuropathy should be sensory demyelinating neuropathy
In the results authors report that neurophysiology in some patients were consistent with predominantly sensory demyelinating neuropathy where motor nerve conduction velocities were slow.
it is unclear why patients had auditory evoked potentials.
Perhaps the results could be presented in a more concise way including a table to show the neurophysiological and clinical features to highlight the clinical variability.
Also in the results patient III-3 does not seem to correspond to the same patient in figure 1.
The discussion is a repetition of the results.
Author Response
Thank you for your comments;
we've corrected the part of Results that were unclear:
"confirming a sensory-motor demyelinating chronic polyneuropathy with a predominant sensory involvment". line 107-108
And we've summarised clinical data in table1 (uploaded)
We've corrected the results referred to patient III-4 (line 122)
we've insert the discussion, in the last submission there was an error, sorry for that!
3. Discussion
Peripheral myelin protein 2 (PMP2) is a small protein of 132 aminoacids located on the cytoplasmatic side of peripheral nervous system (PNS) compact myelin. Nowadays, in literature are reported six PMP2 mutations, clinically correlated to demyelinating neurophaty (CMT1G); the common molecular mechanism seems to alter the function of the protein PMP2, keeping its structure preserved. (Ruskamo et al. 2018, Geroldi et al. 2020).
Although CMT is described as a disease of the peripheral nervous system, CNS involvement is reported in the literature in several CMT patients by variations in the MFN2, PMP22, GJB1, GDAP, MORC-2, NDRG1, and NEFL genes. CNS abnormalities can involve both supratentorial and infratentorial areas: both white matter and grey matter abnormalities, cerebellar abnormalities are also possible. In particular, in the context of demyelinating CMT1, a reduction in bank substance with signs of diffuse hypomyelination with U-fibre sparing has previously been described in several adult patients with CMT1A and HNPP. (Chanson JB et al., 2013) Clinically, these may be asymptomatic or manifest with symptoms including pyramidal signs, optic atrophy, cognitive impairment, cerebellar ataxia, dysphagia, dysarthria, vertigo, ptosis, tremor, sensorineural hearing loss, prolonged visual evoked response, transient sensory impairment. (Lee M, et al., 2017) In all the cases described here a possible CNS involvement was ruled out not only by clinical evaluation but also by evoked potentials and neuroradiological study.
Recently our variant has been described in four members of a family, associated with a clinical and electrophysiological features compatible with a classical CMT1 phenotype with delayed acquisition of motor milestones, areflexia, pes equino, toe walking, frequent falls, distal weakness and muscle hypotrophy and reduced nerve conduction velocities. Clinical features of our family are similar to what previously described for the variant (p.ile50del) by Gerlodi et al., as well as the electronuero-physiological data, which in our sample revealed pictures of sensory-motor neuropathies but with a prevalent sensory involvement (in most cases, in fact, we found reduced motor conduction velocities of less than 35 m/s and absence of potentials on the sensory nerves investigated)
As in our cases, in the majority of cases, the lower limbs are more involved than the upper limbs. (Fusco et al, 2009) Sensory nerve involvement is often less severe than motor nerve involvement; it manifests itself with reduced tactile and vibratory sensitivity or reduced proprioception, which contributes to the onset of the walking disorder. The involvement of the upper extremities leads to a progressive hypotrophy of the intrinsic musculature with possible contractures in flexion of the fingers. Scoliosis is reported in 30-50% of cases, usually with kyphoscoliosis. (Karol, Lori A. et al.,2007).
We here report the same variant described by Geroldi et al. in our index case; this deletion segregates in eight affected members of a three generation CMT1 family. Our patients reported also other 2 family members with gait disturbance and likely distal leg deformity but it was not possible extending them clinical and genetic analysis. An interesting clinical feature in our patients is the presence of early distal tremor in younger patients, which is absent in parents. This intrafamilial phenotypic variability makes the phenomenon of disease anticipation evident. The same genetic mutation led to variable expression within the phenotype, with variation in clinical features and progressively earlier age of onset in successive generations.
Although the pathogenetic mechanisms of the phenomenon of disease anticipation to date are unknown, the hypothesis is that anticipation and intra-family phenotypic variability could be due to the expression of unknown modulator genes or environmental factors, or both. (Fusco C. et al, 2009; Kovach MJ et al, 2002)
In the context of CMT, it is known that the reduction in the expression of connectin 32, which underlies x-linked CMT, may be the result of either an alteration of transcription factors or a mutation of noncoding (P2-promotor) genes. (Houlden H et al, 2004)
The same phenomenon of anticipation associated with a progressive increase in clinical severity has also been described in a subgroup of CMT1A patients. (Steiner I. et al, 2008)
4. Conclusions
Our study describes the clinical features of a family affected by CMT1G from PMP2. The cases re-ported belong to three different generations, giving us the possibility to indirectly observe the evolution of the disease over time and to hypothesise, despite the described intrafamily phenotypic variability and the phenomenon of disease anticipation in younger generation, a benign course of the disease. Interestingly in the adult affected members of the family, a slowly progressive clinical course is observed, with a predominant sensory involvement, associated with muscle atrophy almost exclusively of the intrinsic muscles of the hands and feet, which to date has not led to severe disabilities and without a severe disruption of their quality of life. In the proband sensory nerves are less compromised but it could be correlated to an early stage of the disease, as described in other studies (Geroldi et al 2020, Palaima et al, 2019). Continued follow up of new generations appears necessary to monitor the disease and to confirm the overall benign trend. The possibility of an early diagnosis (clinical, electrophysiological and genetic) allows access to specialist follow-up and therapies, to date only supportive and preventive of the most severe complications, improving the quality of life of patients.

Reviewer 2 Report (New Reviewer)
In this paper, the doctors report a case of CMT disease caused by mutation of PMP2 gene. This gene has been reported and they found another mutation in these family. This will be important to study the PMP2 protein mutation relative CMT disease.
The Figure should redraw and make is more clearly, use high resolution image.
The electrophysiology data should shown in the main text.
Both the grandmother’s mother (I-1) and brother (II-2) showed clinical features of neuropathy with pes cavus and gait disturbance, but no extensive clinical, neurophysiological and genetical data were available. does their grandmother’s mother (I-1)also a mutation in PMP2?
Minor edit:
line 46, delivery should remove the "-".
line 57, "develop-ment" use "development". please check the full text and try to remove the "-" between one word.
line 73, "areflex-ia", the "-" should removed
line 74, "re-duced" use "reduced".
line 78, "hypotrophy" between the characters should remove "-".
line 82, “da-ta" should use "data".
line 95, "mo-tor" should use "motor".
line 108, "domi-nant" use "dominant".
Line 109, 120, 131, 160, 161, 169, 174; please also remove "-", please carefully check all the text.
Author Response
Thank you for your comment.
- we've uploaded a new figure with high resolution, let me now if it's correct
- we've summarised the results in table 1 (uploaded)
- unfortunatly patient I-1 clinical data were not avaiable so we've only the familiar clinical history
- we've corrected every errors and hyphenated word.

Reviewer 3 Report (New Reviewer)
The manuscript entitle "A novel family with demyelinating Charcot Marie Tooth disease by a mutation on the PMP2 gene. Case series of nine patients and a brief review of literature", showed a large family which was referred for a demyelinating motor-sensory neuropathy with autosomal dominant transmission.
Althougth, the articles brought an interesting case of autosomal dominant transmission, the manuscript need be improved.
I notice many hyphenated words, ie, domi-nant and de-livery. I suggest to the authors to review the manuscript to correct these words.
In the discussion, the authors only rewrote the results and didn't do the brief review of literature.
Author Response
thank you for your comments!
- we've corrected all the hyphenated words
- we upload the discussion, in the last version there was an error, sorry for that!
3. Discussion
Peripheral myelin protein 2 (PMP2) is a small protein of 132 aminoacids located on the cytoplasmatic side of peripheral nervous system (PNS) compact myelin. Nowadays, in literature are reported six PMP2 mutations, clinically correlated to demyelinating neurophaty (CMT1G); the common molecular mechanism seems to alter the function of the protein PMP2, keeping its structure preserved. (Ruskamo et al. 2018, Geroldi et al. 2020).
Although CMT is described as a disease of the peripheral nervous system, CNS involvement is reported in the literature in several CMT patients by variations in the MFN2, PMP22, GJB1, GDAP, MORC-2, NDRG1, and NEFL genes. CNS abnormalities can involve both supratentorial and infratentorial areas: both white matter and grey matter abnormalities, cerebellar abnormalities are also possible. In particular, in the context of demyelinating CMT1, a reduction in bank substance with signs of diffuse hypomyelination with U-fibre sparing has previously been described in several adult patients with CMT1A and HNPP. (Chanson JB et al., 2013) Clinically, these may be asymptomatic or manifest with symptoms including pyramidal signs, optic atrophy, cognitive impairment, cerebellar ataxia, dysphagia, dysarthria, vertigo, ptosis, tremor, sensorineural hearing loss, prolonged visual evoked response, transient sensory impairment. (Lee M, et al., 2017) In all the cases described here a possible CNS involvement was ruled out not only by clinical evaluation but also by evoked potentials and neuroradiological study.
Recently our variant has been described in four members of a family, associated with a clinical and electrophysiological features compatible with a classical CMT1 phenotype with delayed acquisition of motor milestones, areflexia, pes equino, toe walking, frequent falls, distal weakness and muscle hypotrophy and reduced nerve conduction velocities. Clinical features of our family are similar to what previously described for the variant (p.ile50del) by Gerlodi et al., as well as the electronuero-physiological data, which in our sample revealed pictures of sensory-motor neuropathies but with a prevalent sensory involvement (in most cases, in fact, we found reduced motor conduction velocities of less than 35 m/s and absence of potentials on the sensory nerves investigated)
As in our cases, in the majority of cases, the lower limbs are more involved than the upper limbs. (Fusco et al, 2009) Sensory nerve involvement is often less severe than motor nerve involvement; it manifests itself with reduced tactile and vibratory sensitivity or reduced proprioception, which contributes to the onset of the walking disorder. The involvement of the upper extremities leads to a progressive hypotrophy of the intrinsic musculature with possible contractures in flexion of the fingers. Scoliosis is reported in 30-50% of cases, usually with kyphoscoliosis. (Karol, Lori A. et al.,2007).
We here report the same variant described by Geroldi et al. in our index case; this deletion segregates in eight affected members of a three generation CMT1 family. Our patients reported also other 2 family members with gait disturbance and likely distal leg deformity but it was not possible extending them clinical and genetic analysis. An interesting clinical feature in our patients is the presence of early distal tremor in younger patients, which is absent in parents. This intrafamilial phenotypic variability makes the phenomenon of disease anticipation evident. The same genetic mutation led to variable expression within the phenotype, with variation in clinical features and progressively earlier age of onset in successive generations.
Although the pathogenetic mechanisms of the phenomenon of disease anticipation to date are unknown, the hypothesis is that anticipation and intra-family phenotypic variability could be due to the expression of unknown modulator genes or environmental factors, or both. (Fusco C. et al, 2009; Kovach MJ et al, 2002)
In the context of CMT, it is known that the reduction in the expression of connectin 32, which underlies x-linked CMT, may be the result of either an alteration of transcription factors or a mutation of noncoding (P2-promotor) genes. (Houlden H et al, 2004)
The same phenomenon of anticipation associated with a progressive increase in clinical severity has also been described in a subgroup of CMT1A patients. (Steiner I. et al, 2008)4. Conclusions
Our study describes the clinical features of a family affected by CMT1G from PMP2. The cases re-ported belong to three different generations, giving us the possibility to indirectly observe the evolution of the disease over time and to hypothesise, despite the described intrafamily phenotypic variability and the phenomenon of disease anticipation in younger generation, a benign course of the disease. Interestingly in the adult affected members of the family, a slowly progressive clinical course is observed, with a predominant sensory involvement, associated with muscle atrophy almost exclusively of the intrinsic muscles of the hands and feet, which to date has not led to severe disabilities and without a severe disruption of their quality of life. In the proband sensory nerves are less compromised but it could be correlated to an early stage of the disease, as described in other studies (Geroldi et al 2020, Palaima et al, 2019). Continued follow up of new generations appears necessary to monitor the disease and to confirm the overall benign trend. The possibility of an early diagnosis (clinical, electrophysiological and genetic) allows access to specialist follow-up and therapies, to date only supportive and preventive of the most severe complications, improving the quality of life of patients.
Reviewer 4 Report (New Reviewer)
Attached.

Author Response
Thank you for ypur comments!
- we add information about PMP2 in the introduction as you've suggested and presented clinical data in table 1 (here uploaded)
1. Introduction
Charcot-Marie-Tooth (CMT) disease represents a genetically and phenotypically highly heterogeneous group of inherited peripheral neuropathies. CMT is classified into demyelinating type (CMT1) with slowed nerve conduction velocity (NCV) below 35 m/s, axonal type (CMT2) with normal or slightly reduced NCV (> 45 m/s), and intermediate type (I-CMT) with signs of both demyelination and axonal degeneration and NCVs between 35 and 45 m/s.
The exponential increase in the number of known genes associated with CMT pictures, possible mainly thanks to exome sequencing (WES) techniques, has made traditional classification systems increasingly complex, leading since 2018 to a revision of these systems and classification based on causative gene alterations (Magy et al., 2018). Within each category (CMT1, CMT2...), the association with a specific gene is indicated by a letter of the alphabet (e.g. CMT1A, CMT1B...).
CMTs today represent a broad genetic and clinical spectrum; it is not uncommon to find several mutations on the same gene associated with different phenotypes and, likewise, it is possible for the involvement of different genes to be associated with the same clinical phenotype.
Genetic variability manifests itself both in different inheritance patterns (autosomal dominant, autosomal recessive, X-linked) and in distinct electroneurophysiological classes (demyelinating, axonal and dominant intermediate). Autosomal dominant forms are the most frequent; a few sporadic cases related to de novo mutations have been described (less than 10% of CMT1A). (Morena et. Al, 2019)
Currently, more than 90 different pathogenic mutations are reported in the literature (Pisciotta C. et al, 2018) involving protein synthesis and post-transcriptional regulation processes, intracellular transport or mitochondrial function. (Morena J. et al., 2019) 90% of cases are associated with a variation in four genes: PMP22, MPZ, GJB1 and MFN2, in particular about 50% of CMT cases are associated with duplication/deletion of the PMP22 gene (Bird TD. et al., 2021). To date, the genetic cause is still unknown in 20-30% of hereditary neuropathies, but this percentage is gradually decreasing thanks to new molecular techniques. (Jani-Acsadi A. et al., 2015). CMT1, demyelinating type, accounts for 40–50% of all CMT patients. (Palaima P et al. 2019) Myelin sheath is composed by a high fraction of lipids and myelin specific proteins: myelin basic protein (MBP), myelin protein zero (MPZ), peripheral myelin protein 22 (PMP22) and peripheral myelin protein 2 (PMP2). Any slight alteration in its protein and lipid components alters the integrity of the structure and thus of the axonal lining, resulting in reduced nerve conduction velocities. (Palaima P et al. 2019) The most frequent cause of CMT1 is alterations in PMP22 gene, resulting more frequently in CMT1A, less often in HNPP and CMT1E, followed by MPZ mutations which lead to CMT1B. MBP has not been implicated in CMT so far. PMP2 has recently emerged as a novel rare cause of dominant CMT1. (Gonzaga-Jauregui C. et al., 2015, Hong YB et al. 2016, Motley WW et al. 2016, Palaima P et al. 2019, Geroldi et al. 2020). PMP2 is a small 14 kDa protein, located on the cytoplasmic side of compact myelin, belonging to the fatty acid binding protein family (FABP) and it is characterized by an antiparallel β-Barrel structure with an α-helical-cap. (Palaima P et al. 2019, Gerlodi et al, 2020) Its function in the peripheral nervous system remains partially unclear, but a role in the lipidic homeostasis is suggested from the literature. Stettner et al. also suggested a PMP2 role in the remyelination process of the injured peripheral nervous system. (Stettner 2017)
The typical clinical spectrum is characterized by distal weakness, sensory loss, foot deformities and absence of reflexes. In the CMT1 forms reported in literature progressively appear distal hyposthenia of the feet and hands with muscle atrophy. Slowly progressive sensory involvement with loss of proprioception and vibratory sensitivity is possible. In young adulthood there is atrophy of the hand and foot muscles to a greater degree, possible hypertrophy of the nerves, palpable, and kyphosis-scoliosis-like deformities of the spine. The evolution is benign in most cases, cases of loss of ambulation described. Worsening in pregnancy is also described.
Here we report clinical and electrophysiological data from of all the members for three generations of a CMT family harboring a heterozygous deletion c.147_149delTAT in PMP2 gene which lead to an isoleucine loss at codon 50 (p.Ile50del) that segregates with the disease, previously described in the literature by Gerlodi et al. (Gerlodi et al, 2020).

Reviewer 5 Report (New Reviewer)
The authors presented an interesting manuscript describing a large family with CMT1G. Despite representing one of the classic autosomal dominant demyelinating CMT, PMP2 variants have been scarcely studied in some contexts, such as in adult patients and in aspects related to the natural history of motor and sensory compromise. Some points can be evaluated at this time by the authors to improve the quality of their manuscript:
1. The Abstract description does not give a proper idea that a large family with CMT1G is going to be studied in detail. I suggest a review of aspects of the structure of the text in the Abstract to make it easier to understand for the general reader.
2. Several typos are present in the text and should be carefully reviewed and corrected by the authors. Some of them: line 25: "phenotipically" => phenotypically; line 41: "codone 50" => codon 50; line 117: "analized" => analyzed; line 130: "undetactable" => undetectable; line 136: "muscolar wasting" => muscular wasting; line 175: "quality life" => quality of life.
Author Response
Thank you for your comments!
- we've reviewed the abstract as suggested:
Abstract: Introduction: Charcot-Marie-Tooth (CMT) are a group of inherited peripheral neuropathies characterized by wide genotypic and phenotypic variability. The onset is typically in childhood, and the most frequent clinical manifestations are predominantly distal muscle weakness, hypoesthesia, foot deformity (pes cavus), and areflexia. In the long term, complications such as muscle-tendon retractions, extremity deformities, muscle atrophy, and pain may occur. Among CMT1, demyelinating and autosomal dominant forms, CMT1G is determined by mutations in the PMP2 myelin protein. Results: starting from the index case, we performed clinical, electrophysiological, neuroradiological and genetic evaluation of all family members for three generations; we identified p.Ile50del in PMP2 in all the nine affected members; they presented a typical clinical phenotype, with childhood onset, variable severity between generations and a chronic demyelinating sensory-motor polyneuropathy at the electrophysiologic examination, slow to very slow progression, predominant at the lower limbs. Our study reports a relatively large sample of patients with CMT1G by PMP2, a rare form of demyilinating CMT, highlighting the genetic variability of the CMT family instead the overlapping clinical phenotypes within demyelinating forms. In addition, we believe that the early diagnosis (clinical, electrophysiological and genetic) allows access to specialist follow-up and therapies, to date only supportive and preventive of the most severe complications, improving the quality of life of patients.
- we've corrected what you've highlighted in the text
Round 2
Reviewer 1 Report (Previous Reviewer 4)
The authors have reviewed the manuscript and added in the abstract some conclusion. In the conclusion they report that their study reports a relative large sample of patients with CMT1G by PMP2. I would suggest to change in to one family as patients are members of the same family.
They also include a long paragraph in the introduction regarding classification of CMT but I am not sure whether this was requested and it was not clearly highlighted in the response to the reviewers.
They included a table summarising the clinical features which is better as the results section is too long.
It is unclear why the authors comment on CNS involvement and arranged MRI brain and auditory evoked potentials as the patients reported did not present any signs of CNS involvement and in previous reports no additional distinguishing features were reported. Therefore I do not think the comment on the CNS involvement is relevant in this particular family.
In addition the authors did not highlight how their family differs from the one already reported by Geroldi et al and the intrafamilial variability is not completely apparent apart from the tremor starting slightly earlier but younger patients do not seem to be more severely affected than their parent.
Author Response
Thank you for your suggestions.
-We have corrected the abstract by specifying that the cases described are the members of a single family.
- we decided to include the introductory part on CMT to complement the brief literature review on PMP2 cases described so far
- as suggested we have instead lightened the reference to central nervous system involvement; our patients were subjected to in-depth diagnostic investigations in the light of the initial negative results of the first genetic panels performed and to complete a differential diagnosis.
- we have better emphasised in the discussion the early sensory involvement found in our patients compared to the sample described by Geroldi and how, compared to the still few cases described in the literature, the presence of anticipatory tremor is significant. Also significant is the evolutionary mirror of the disease provided by the wide age range of our sample.
Reviewer 3 Report (New Reviewer)
I would like to thank the authors for address my questions and concerns.
I strongly believe that new version of the article is improved, however it needs a text editing before to be publish.
The authors need to pay attention to the reference on the text, for example, in a paragraph that starts on line 173 I can notice punctuation errors.
Author Response
Thamk you for your comments,
We've corrected the references in the text.
Reviewer 4 Report (New Reviewer)
Most of the concerns were addressed. Thanks
Author Response
thank you so much for your comments and for your help
This manuscript is a resubmission of an earlier submission. The following is a list of the peer review reports and author responses from that submission.
Round 1
Reviewer 1 Report
The authors report a family harboring a deletion 94 c.147_149delTAT in the PMP2 gene leading to an isoleucine loss at codone 50 (p.Ile50del). The mutation generated a demyelinating Charcot Marie Tooth Type-1G phenotype. The findings are largely consistent with a previous report on a different family with the same mutation (Gerlodi et al. 2020).
The authors conclude the novelty of the study by emphasizing that “this is the longest follow up of CMT1G from PMP2 reported that shows, despite the intra-family phenotypic variability, a benign course of the disease”.
The report is valuable and objectively written, nevertheless the discussion is unexpectedly brief.
For clinical relevance, the authors could discuss the clinical/electrodiagnostic features that allow to differentiate the PMP2 deletion from other CMT1 neuropathies.
For scientific relevance, it could be relevant to discuss the potential causes of the phenotypic variability between the family beyond age.
Minor:
- In the abstract it is superfluous to review the electrophysiological CMT classification in general. Focus on the electrophysiological characterization of the family.
- what do the authors mean by ”it was not possible extending them clinical and genetic analysis”.
Author Response
Response 1: Dear auditor, thank you for your suggestions;
- as recommended, we have enriched the discussion of the case report by also expanding on the aspect of phenotypic variability described;
- as stated in the text, according to the knowledge we have today, the clinic of CMT1G is overlapping with the more common CMT1A.
- we have modified the abstract leaving only the relevant information concerning the family described, as suggested;
- we have explained in the text that the information concerning the patients referred to in the sentence is only anmanestic as they are deceased subjects.
Reviewer 2 Report
The authors present a case study of an extensive family with a CMT1 phenotype cause by a previously reported PMP2 variant. This manuscript whilst very basic and not expanding much on known literature relating to PMP2 neuropathies, reports like this are important for estimating prevalence and severity of phenotypes. While, the study is well performed, additional details and changes in the presentation could make this a qualitatively better manuscript.
Abstract:
Line 24-26: Dominant missense variants in PMP2 were first described in 2015, which is not exactly recent anymore.
Introduction:
Line 43-44: HNPP is indeed a neuropathy, but not a subdivision of CMT1, which is implied by this sentence.
Line 47-48: It would be good to mention the specific mutation here and report that this is a known mutation already.
Case Report:
Line 57: "NCV studies of four limbs showed reduced motor and sensory velocities.", and normal amplitudes or were these also already diminished. Also, please quantify more clearly the extend of the loss of nerve conduction velocity, by either specifying the actual speed or mild/moderate/severe.
It would be beneficial to provide a table with the clinical characteristics of all the described patients. This would be especially pressing as the authors claim that this is the longest follow-up of a PMP2 family (Conclusion - Line 120). So far, it seems as no longitudinal data has been presented directly countering the "longest follow-up" argument.
Line 66-67: It would be good to pick a position in whether this grandmother has the same phenotype, as the authors state that while she shows pes cavus, areflexia and muscle atrophy, they also state that there is no severe neuromuscular issue. All three symptoms described as present could be something related to age rather than CMT, was the grandmother tested for the variant and if so, is she a carrier?
Line 96-97: "Segregation analysis confirmed the presence of this deletion in all the affected family 96 members." This is by definition untrue, as a previous sentence says that some individuals were not available for testing. It would be important to specify genotypes of affected and unaffected in the pedigree and change this sentence to "Segregation analysis confirmed the presence of this deletion in all available affected family members."
Figures:
A significant improvement is needed for the Figure. The pedigree figure looks messy and some numbers and parts of the text are not legible. The numbering should include all individuals depicted, not just those discussed in the text. It is essential to specify genotypes in the pedigree as well as specify which individuals have been clinically examined and which individuals are hearsay affected or hearsay unaffected. The "Legenda" should be in English, so "Legend".
Minor remarks:
- Check the manuscript for double spaces as in Line 52
Author Response
Response 2:
Dear reviewer, thank you for your comments. We have added some interesting clinical details and extended the discussion.
ABSTRACT
line 24-26 (now 29-30): we have modified the sentence "Among CMT1, demyelinating and autosomal recessive forms, CMT1G, is determined by mutations in the PMP2 myelin protein "
INTRODUCTION
line 43-44 (now 46-47): "The most frequent cause of CMT1 is alterations in PMP22 gene, resulting more frequently in CMT1A, less often in HNPP and CMT1E"
line 47-48 (now50-51): we have mentioned the specific mutation as suggested
CASE REPORT
line 57 (now 62-63): we have added neurophysiolocical data in the text and modified the conclusion try to explain better our intent (line 144-147 "The cases reported belong to three different generations, giving us the possibility to indirectly observe the evolution of the disease over time and to hypothesise, despite the described intra-family phenotypic variability and the phenomenon of disease anticipation in younger generation, a benign course of the disease" and line 150-151 "Continued follow-up of new generations appears necessary to monitor the disease and to confirm the overall benign trend."
line 66-67: we have highlighted in line 108 and in the figure all patients genotype.
line 96-97 (now 107-108): we have modiefied the sentence as suggested
FIGURE
we have completed the figure and modified the legend as suggested
Reviewer 3 Report
The authors reported a novel family with demyelinating Charcot Marie Tooth dis-2 ease caused by a mutation of the PMP2 gene. It is an interesting case report, but few novelty can be found in the study. Also only the simple clinical and eletrophysiological features were described in the study. Some phrases are difficult to understand such as conduction velocity values (NVC).
Author Response
Response 3:
Thank you for your comment; as suggested, we have deepened some further clinical aspects and expanded the discussion on this issue.
Reviewer 4 Report
Baga et al present a four generation family with autosomal dominant history of typical demyelinating Charcot Marie Tooth disease due to already published PMP2 mutation. Although the case report is interesting as the mutation segregates in all affected family members it does not add much compared to the previous publication by Geroldi et al. In the discussion no clear difference compared to the other family reported by Geroldi et al is highlighted.
Moreover some stylistic points should be addressed: in the abstract the definition of demyelinating CMT is reported to be <35 m/s and axonal >45 m/s on neurophysiology whereas in the introduction it is clearly defined as demyelinating with CV <38 m/s, Axonal >38m/s and intermediate between 35-45 m/s.
Also in the introduction only CMT1A due to PMP22 duplication should be considered the most common but not HNPP or CMT due to PMP22 point mutation.
In the case report it would be important to include the conduction velocities in order to show the neurophysiological features of the neuropathy.
the past tense should be also used throughout the case report.
line 55 were should be used instead of was
line 56 abolished deep tendon reflexes should be used instead of deep tendon reflexes abolished
line 68 have been observed should be used instead of has been observed
also the sentence severe distal weakness with no muscle trophism abnormalities is not clear. Does it mean that there was no wasting despite weakness?
Author Response
Response 4 :
- we have corrected the definition of demyelinating CMT in the discussion and remove it in the abstract as suggest by another reviewer. "CMT is classified into demyelinating type (CMT1) with slowed nerve conduction velocity (NCV) below 35 m/s, axonal type (CMT2) with normal or slightly reduced NCV (> 45 m/s), and intermediate type (I-CMT) with signs of both demyelination and axonal degeneration and NCVs between 35 and 45 m/s."
- as suggested we modified the introduction "The most frequent cause of CMT1 is alterations in PMP22 gene, resulting more frequently in CMT1A, less often in HNPP and CMT1E, followed by MPZ mutations which lead to CMT1B."
- we have added in the text avaiable neurophysiologic data
- line 55 (now 60): we have modified with "should be used"
- line 56(now 62) : we have modified with "abolished deep tendon reflexes"
- line 68(now 81): we have modified with "have been observed" and explain the sentence about muscle weakness "no muscolar wasting despite severe distal weakness."